# Survey for Major Grapevine Viruses in Commercial Vineyards of Northwestern Argentina

**DOI:** 10.3390/plants11131720

**Published:** 2022-06-28

**Authors:** Mónica Rivadeneira, Marta Zulema Galván, Marina Abán, Rosa Elena Semke, Josefina Rivadeneira, Melisa Lanza Volpe, Sebastián Gomez Talquenca

**Affiliations:** 1Instituto Nacional de Tecnología Agropecuaria (INTA) EEA Salta, Ruta Nacional 68 Km 172 (4403) Cerrillos, Salta 4403, Argentina; galvan.marta@inta.gob.ar (M.Z.G.); marp.aban@gmail.com (M.A.); rivadeneira.josefina@inta.gob.ar (J.R.); 2Consejo Nacional de Investigaciones Científicas y Técnicas (CONICET) CCT-Salta, J.M. Leguizamón 366, Salta 4400, Argentina; 3Centro de Desarrollo Vitícola del Valle Calchaquí, Cafayate, Salta 4427, Argentina; rosasemke495@hotmail.com; 4Instituto Nacional de Tecnología Agropecuaria (INTA) EEA, Mendoza 5602, Argentina; lanzavolpe.melisa@inta.gob.ar

**Keywords:** grapevine viruses, *Vitis vinifera*, Calchaquíes Valleys

## Abstract

This study aimed to survey the occurrence of eight grapevine viruses in commercial vineyards located in the Calchaquíes Valleys in the northwest region of Argentina. A total of 103 samples of mature canes of vines showing either none or some viral-like symptoms were randomly collected. The samples were tested by RT-PCR/PCR-based assays for the screening of the following viruses: *Grapevine fanleaf virus* (GFLV), *Grapevine leafroll-associated viruses* (GLRaV-1, -2, -3, -4), *Grapevine virus A* (GVA), *Grapevine rupestris stem pitting-associated viruses* (GRSPaV), and *Grapevine red blotch virus* (GRBV). Sixty percent of the analyzed samples showed infection with some of the analyzed viruses, except GRBV. GLRaV-3 and GFLV were the most frequent viruses, present in 34% and 21% of the positive samples, respectively. This study represents the first survey report of the presence of grapevine viruses in the region of the Calchaquíes Valleys and contributes to the knowledge to maintain the sanitary status of commercial vineyards in Argentina.

## 1. Introduction

Grapevine (*Vitis vinifera*) is the main fruit crop in Argentina. According to the International Organization of Vine and Wine [1], Argentina is the seventh largest grapevine producer worldwide with 218 thousand hectares, and its vineyards are considered the highest and southernmost in the world. The northwestern region of Argentina represents the second largest wine region of the country, after the Cuyo region, with 6511 hectares located in the Calchaquíes Valleys, an area that includes Salta, Tucumán, and Catamarca provinces [2]. In these valleys, red grapes represent 66% of the surface planted with grapevine cultivars, with the Malbec and Cabernet Sauvignon being the prevailing varieties, whereas white grapes represent 32% and pink grapes 2%.

In the last 20 years, the cultivated area in the Calchaquíes Valleys has increased by 92%, with a considerable increase in the number of both commercially large vineyards (158%) and small artisanal producers (1767%). The trend is that the cultivated area, as well as the expansion and establishment of vineyards, will continue to increase. Most of the planting materials (either certified virus-free or not) used to establish vineyards in the Calchaquíes Valleys are imported from nurseries located in the provinces of Mendoza and San Juan, in the Cuyo region of Argentina, with a smaller proportion coming from foreign nurseries or being generated by the producer themselves. Thus, the competitiveness of the Calchaquíes Valleys wine sector depends, in part, on the performance of the planted vines, for which it is necessary to be certain of their quality and health aspects.

Grapevine cultivars are commonly affected by a wide range of biotic factors, among which viruses are considered the most threatening due to the lack of therapeutic methods. More than 80 distinct grapevine viruses have been identified worldwide [3]. All these viruses are disseminated by means of infected propagation material and lead to considerable decreases in fruit quality and yield, causing significant economic losses for producers and nurserymen [4,5,6].

The occurrence of Grapevine leafroll-associated virus (GLRaV)-1, GLRaV-2, GLRaV-3, GLRaV-4, Grapevine fanleaf virus (GFLV), Grapevine virus A (GVA), Grapevine rupestris stem pitting-associated virus (GRSPaV), and Grapevine red blotch virus (GRBV) has been identified and reported in vineyards in the province of Mendoza, with yield losses in most of the cases [7,8,9,10]. However, the presence or characterization of viral diseases in cultivars in the northwest region of the country has not been assessed before. As the most common method of grapevine reproduction is agamic propagation through dormant cuttings, favored if the original plant material is infected, long-distance dissemination of viruses is favored. Considering that most of the grapevine planting materials come from the Cuyo region, it is possible that the viruses reported there are present in the Calchaquíes Valleys vineyards as well. In addition, the presence of some virus vectors, such as the vine mealybugs (Planoccous ficus), has been reported in some locations of the Calchaquíes Valleys [11] and considering its role in the natural dispersion of GLRaV-3 in Argentina [12], there is a high risk for virus spread from neighboring locations within the Calchaquíes Valleys.

Due to the variable symptomatology observed between and within cultivars, which hinders the visual diagnosis, molecular diagnosis becomes necessary for accurate identification of the etiology of viral diseases, as well as for their subsequent control and management [13,14]. Thereby, the aim of this study was to diagnose the spectrum of viruses present in vineyards of the northwestern region of Argentina, through field surveys and molecular testing to generate knowledge on the presence of viruses to implement sanitation and management practices. The study was conducted in the localities of Cachi, Molinos, San Carlos, and Cafayate in Salta province and Tafí del Valle in Tucumán province throughout the 2019 to 2020 seasons (Figure 1).

## 2. Results and Discussion

Our results demonstrated that vineyards in the Calchaquíes Valleys are infected by, at least, seven of the most common grapevine viruses in the Cuyo region and worldwide (GFLV, GLRaV-1, -2, -3, -4, GVA, and GRSPaV) as GRBV was not detected in this study. From the 103 randomly selected samples, 61 (59%) tested positive for at least one virus (Appendix A). Single infections of each virus were found in 37 samples (36%). The most prevalent virus was GLRaV-3, present in 34% of the total samples, which was expected, as it is considered one of the most widely dispersed grapevine viruses worldwide [15]. This virus has also been reported as one of the most prevalent in Mendoza province [8], from where most of the planting materials of the Calchaquíes Valleys are commonly imported. GLRaV-3, which was identified in 35 samples, presented twelve different mixed infections. The GLRaV-3/GVA and GLRaV-3/GFLV combinations were the most common (Figure 2). There were no cases of mixed infections of more than two different GLRaVs together. GLRaV-3 was the only GLRaV present in mixed infections with GLRaV-1, GLRaV-2, or GLRaV-4. The common co-occurrence of GVA and GLRaVs in the samples tested agree with the conclusions of Rowhani and colleagues [16], who observed that the occurrence of Vitiviruses such as GVA were unusual in the absence of mixed with leafroll viruses.

Cafayate was the locality with the highest prevalence of GLRaV-3 (100%), possibly due to the presence of mealybugs (*P. ficus*) observed only in this locality (Table 1). The presence of this vector in the southern region of the Calchaquíes Valleys is in agreement with previous reports [11] and would also explain the highest prevalence of GLRaV-1 in Cafayate. Many reports have shown GLRaV transmission by *P. ficus* [17]. However, due to the limited mobility of female mealybugs, which are the ones with the ability to transmit viruses, the virus spread within a vineyard is slow unless the insect is dispersed by other means, such as human activities (carried on workers’ clothing or harvesting equipment), ants, wind-blown infested leaves, or foraging birds [13].

GLRaV-3 was also detected in localities where mealybugs were not found: San Carlos (60%), Cachi (22%), Tafí del Valle (14%), and Molinos (4%) (Table 1). Considering that GLRaVs are not mechanically transmissible between grapevines (by pruning shears, trimmers, harvesters, or saws) [13], the high GLRaVs prevalence observed might be explained by the introduction of infected propagation plant material in these localities. Thus, establishing new vineyards with virus-tested certified planting material constitutes an essential strategy for the control of leafroll disease in these locations.

GFLV was the second most frequent virus, present in 22 samples (21%), half of them in mixed infections with other viruses detected. The localities with the highest prevalence of GFLV were Cafayate and Tafí del Valle (27%), followed by San Carlos (20%), Cachi (19%), and Molinos (15%). GFLV is one of the most widespread and damaging viruses worldwide [18]. These results warn of the need to carry out a thorough research of the presence of its vector, *Xiphinema index*, in the soils of the Calchaquíes Valleys and to determine whether it is contributing to GFLV spread.

Although GRBV was not detected in the vine samples studied, the absence of this virus in the region cannot be confirmed with certainty, considering it has been previously reported in the country, in the province of Mendoza, by Luna and colleagues [10].

Forty-two of the grapevines studied (41%) were negative for the eight viruses tested. This suggests that the symptoms observed in these plants could be due to other viruses not included in the present study, could have a fungal or bacterial origin, or even abiotic disorders.

This survey confirmed the presence of several viral diseases, known to cause major economic losses worldwide, in the grapevines cultivars of the Calchaquíes Valleys. Taking into account that there is no cure for vines with virus infections, there is a need of proper early diagnosis and pathogen identification in the region, especially in propagation planting material, to prevent and/or manage viral diseases [19,20]. Moreover, four out of the nine asymptomatic plants sampled in the present study tested positive for GFLV, GLRaV-4, and/or GVA, suggesting that asymptomatic plants could be a source of inoculum if used as propagation material. The occurrence of asymptomatic infections in grapevine cultivars has been previously reported [21,22].

This study represents the first report of the presence of grapevine viruses in the region of the Calchaquíes Valleys and contributes to the knowledge of the sanitary status in commercial vineyards in Argentina, along with previous reports of viral diseases carried out in the Cuyo region [7,8,10,23]. To achieve further knowledge of the health status of grapevines in the region, we consider it essential to perform thorough studies on virus incidence and genetic diversity, monitoring of vectors, etc., as well as to analyze the economic impact caused by the presence of these viruses.

## 3. Materials and Methods

### 3.1. Sampling Survey

The survey was carried out in the grapevine-growing region of Calchaquíes Valleys, in northwestern Argentina, in the localities of Cachi, Molinos, San Carlos, and Cafayate in Salta province, and Tafí del Valle in Tucumán province, throughout the 2019 to 2020 seasons (Figure 1). Vines expressing viral-like symptoms on leaves, such as malformations, redness, discolorations, yellowing, and leaf-rolling, and some non-symptomatic vines were collected from randomly selected vineyard blocks and processed for the screening of eight viruses: GLRaV -1, -2, -3, -4, GVA, GFLV, GRSPaV, and GRBV. Two mature canes from each branch of 103 vines were sampled, stored in plastic bags, and transported in ice coolers to the laboratory for the molecular analysis. The grapevine cultivars sampled were: Malbec, Criolla chica, Cabernet Sauvignon, Tannat, Petit Verdot, Syrah, Merlot, Torrontes Riojano, Aspirant Bouschet, and Sangiovesse. The presence/absence of mealybugs (*P. ficus*) was also recorded at the end of summer and the beginning of autumn (February and March, respectively). All plant samples used in this study were collected with authorization of the vineyard owners. The vineyards’ names and geographical locations are kept confidential to protect the anonymity of the growers who participated in the study.

### 3.2. Molecular Diagnosis

RNA was extracted following the rapid CTAB-based procedure described by Gambino and colleagues [24]. RNA integrity was assessed from the 28S and 18S rRNA bands on 1% non-denaturing agarose gel electrophoresis, stained with GelRed^TM^ (Genbiotech, Argentina) and visualized under UV light. First-strand cDNA was synthesized as described by Gambino and Gribaudo [25], and the resulting cDNA was subjected to PCR amplification using primers for Vitis 18S rRNA and eight viruses: GLRaV -1, -2, -3, -4, GVA, GFLV, GRSPaV, and GRBV (Table 2) [25,26,27,28,29]. The PCR reaction mix (10 μL) contained 3 μL of cDNA, 0.02 mM of each dNTP, 0.2 μM of each primer, 1.25 mM MgCl_2_, and 0.5 U *Taq* DNA polymerase (Genbiotech, Argentina). The cycling conditions consisted of initial denaturation at 94 °C for 4 min, followed by 35 cycles at 94 °C for 30 s, 50 °C for 60 s, and 72 °C for 90 s and final extension at 72 °C for 5 min. Reaction products were resolved by electrophoresis on 1.5% agarose gels buffered in TAE 1X (45 mM Tris–acetate, 1 mM EDTA) and visualized under UV light after staining with GelRed^TM^ (Genbiotech, Argentina). Positive DNA samples for each virus were included in the PCR reactions as controls.

## 4. Conclusions

This is the first study to identify viruses that affect vines in the Calchaquíes Valleys, the second largest wine region of Argentina. The results found warn of problems in the propagation material used in this region and the need for differentiated management strategies in localities with and without mealybugs. It is imperative to carry out more in-depth studies on the viruses that affect the vine and its vectors in the area.

## Figures and Tables

**Figure 1 plants-11-01720-f001:**
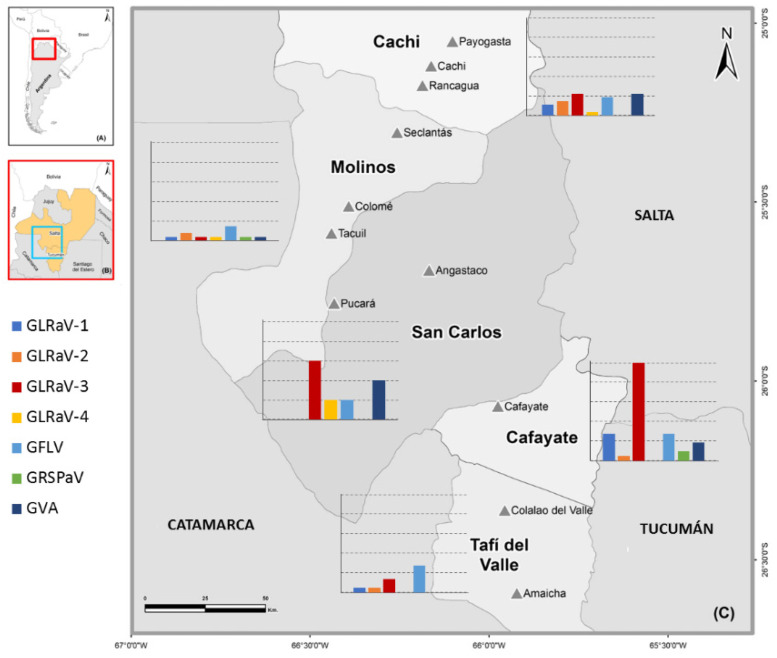
Map showing the sampling localities for the grapevine virus survey in northwestern Argentina. (**A**) Map highlighting the location of northwestern Argentina. (**B**) Map of northwestern Argentina showing sampled provinces. (**C**) Sampled locations (▲) and geographic distribution of viruses detected among localities (bar charts). The viruses shared between sites are represented by the same color. Percentage of positive samples is indicated on the y axis. Dotted lines represent 20% increments.

**Figure 2 plants-11-01720-f002:**
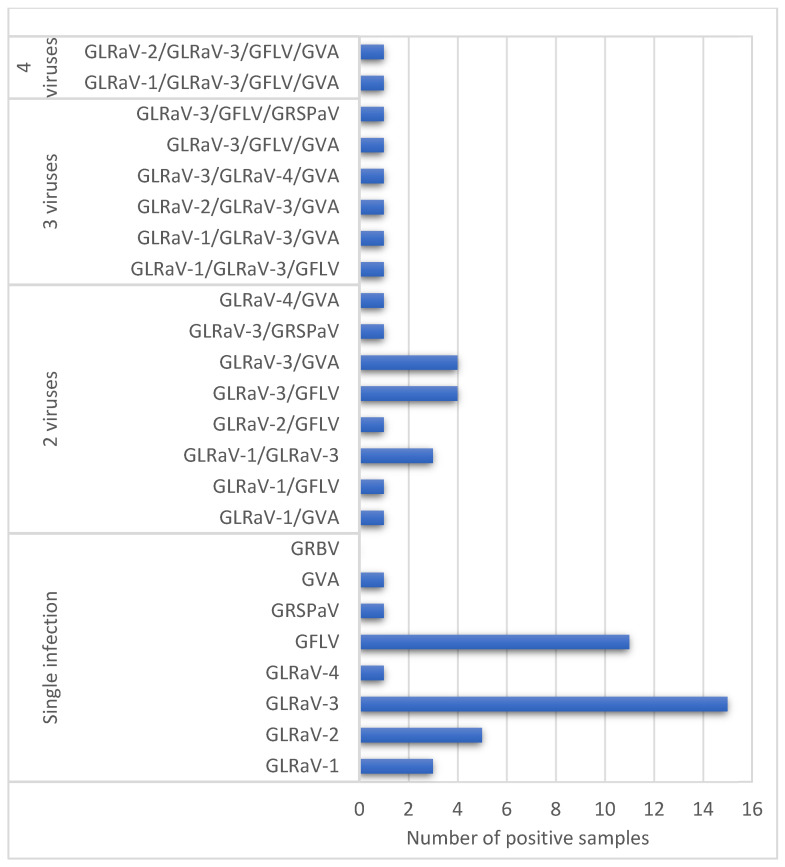
Virus and virus mixed infections detected in the grapevines from northwestern Argentina studied.

**Table 1 plants-11-01720-t001:** Number of positive samples detected for each grapevine virus analyzed by locality.

Location	N ^a^	*P. ficus* ^b^	GLRaV-1 ^c^	GLRaV-2	GLRaV-3	GLRaV-4	GRBV	GFLV	GRSPaV	GVA
Cachi	27	-	3	4	6	1	0	5	0	6
Molinos	27	-	1	2	1	1	0	4	1	1
San Carlos	5	-	0	0	3	1	0	1	0	2
Cafayate	22	+	6	1	22	0	0	6	2	4
Tafí del Valle	22	-	1	1	3	0	0	6	0	0
Total	103		11	8	35	3	0	22	3	13

^a^ Number of samples; ^b^ Presence (+) or absence (-) of Planococus ficus; ^c^ GLRaV: Grapevine leafroll-associated virus, GRBV: Grapevine red blotch virus, GFLV: Grapevine fanleaf virus, GRSPaV: Grapevine rupestris stem pitting-associated virus, and GVA: Grapevine virus A.

**Table 2 plants-11-01720-t002:** Primer sequences used in the RNA analysis for screening of eight grapevine viruses.

Target	Primer sequences (5′-3′)	Location	Product Size (bp)	Gene	Reference
18S rRNA	F: CGCATCATTCAAATTTCTGC R: TTCAGCCTTGCGACCATACT	215-234 1039-1058	844	Internal control	[25]
GLRaV-1	F: TCTTTACCAACCCCGAGATGAA R: GTGTCTGGTGACGTGCTAAACG	7245-7266 7455-7476	232	Coat protein	[25]
GLRaV-2	F: GGTGATAACCGACGCCTCTA R: CCTAGCTGACGCAGATTGCT	6745-6764 7268-7287	543	Coat protein	[25]
GLRaV-3	F: TACGTTAAGGACGGGACACAGG R: TGCGGCATTAATCTTCATTG	13383-13404 13699-13718	336	Coat protein	[25]
GLRaV-4	F: TGAGGTCCCATGTCATGAC R: CCTCAATCTRTTSACCAAYTCAC	7499-7517 7934-7956	457	RNA-dependent RNA polimerase	[26]
GVA	F: GAGGTAGATATAGTAGGACCTA R: TCGAACATAACCTGTGGCTC	6591-6612 6843-6862	272	Coat protein	[27]
GFLV	F: ATGCTGGATATCGTGACCCTGT R: GAAGGTATGCCTGCTTCAGTGG	5506-5527 5602-5623	118	RNA-dependent RNA polimerase	[25]
RSPaV	F: TGAAGGCTTTAGGGGTTAG R: CTTAACCCAGCCTTGAAAT	7708-7726 8612-8593	905	Coat protein	[28]
GRBaV	F: CAAGTCGTTGTAGATTGAGGACGTATTGG R: AGCCACACCTACACGCCTTGCTCATC	2567-2595 2884-2850	318	Replication-associated protein gene fragment (Rep)	[29]

GLRaV: Grapevine leafroll associated virus; GFLV: Grapevine fanleaf virus; GVA: Grapevine virus A; GRBaV: Grapevine red blotch-associated virus; RSPaV: Grapevine rupestris stem pitting-associated viruses. F: Forward primer; and R: Reverse primer.

## Data Availability

Not applicable.

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
