# Peer review of "Survey for Major Grapevine Viruses in Commercial Vineyards of Northwestern Argentina"

_plants, 2022, doi:10.3390/plants11131720_

Round 1
Reviewer 1 Report
In this study authors surveyed a commercial viticultural area for the presence of different grapevine viruses. This is a simple, straight forward paper. Even though the data generated in this study is limited, still it is valuable for improving sanitary status of vineyards in that region. Authors should have considered generating sequence information for some these virus isolates, to have a better understanding of the nature of virus variants distributed in the region. One of the main drawbacks of the manuscript is that the draft requires significant improvement in the language. I pointed some but the draft needs thorough editing.
Following are some minor suggestions
L22: delete "selected and"
L23: modify to "The samples were tested by RT-PCR/PCR based assays for the following viral species: "
L24: delete "They were analyzed by RT-PCR using specific pri-25 mers for each one."
L28: change "first report" to "first field survey report"
L29: change "the knowledge of the sanitary status in commercial vineyards" to "the knowledge to maintain the sanitary status of commercial vineyards"
L35: 8th what? in terms of acreage or wine production
L37: "second wine region"....Is it second largest? be specific
L38: "red vines" not sure if this is right wording
L39-41: rephrase the sentence
L43: "incrementation increase?"
L45: "stabilishing new ones?"
L46: "planting materials (certified virus-free or not) of the" change to ""planting material (certified virus-free or not) used for establishing vineyards in CV are"
L52: "treatments" to "methods"
L72: "to perform an" replace with "for"
L74: "to perform a first diagnosis of grapevines" sounding odd..rephrase using some simple sentences "to diagnose the spectrum of viruses present in vineyards"
L88: "presented twelve different combined infections" ???? rephrase
Use the wording "mixed infections" for co-infection of two or more viruses in one sample.
L112-113: rephrase "GLRaV-3 was also detected in locations where mealybugs were not detected:"
L139: "vines samples" to "vine samples"
L139: delete "we cannot assure"
"Although GRBV was not detected in these vine samples, the absence of this virus in the region can not be concluded with certainty."
L150: You may want to indicate even at the beginning of the results section that nine out of the 103 samples collected were asymptomatic. Also, indicate the viruses that were detected in these asymptomatic samples.
L133-162: so many small paragraphs. with a proper flow try to put it in couple of paragraphs.
L175: Indicate the time of the year (month) you collected the mealybug presence/ absence data.
L194: "second wine region" correct it , second largest ...?
Figure 1: rephrase the legend
"Virus and virus coinfections detected in the grapevines from NorthWestern Argentina"
The primer sequences table (supplementary table S2) can be considered to move into main draft if it is allowed as per short communication instructions.
Author Response
Reviewer suggested: “Authors should have considered generating sequence information for some of these virus isolates, to have a better understanding of the nature of virus variants distributed in the region”.
We agree with the reviewer. In fact, the main goal of our research program is to perform a comprehensive study of the viruses infecting grapevine in Northwestern Argentina. However, the objective of this first work was to identify the occurrence of the viruses in the region, in order to justify a future molecular characterization study based on these results. Currently, we are performing a characterization of the genetic variability of the local isolates through a sanger sequencing approach, but we will be unable to incorporate such results in the present manuscript, as we need to perform an exhaustive cloning and sequencing work to obtain robust results.
As suggested by the reviewer the following changes were made:
L22: "selected and" was deleted
L23: the phrase “The viral species selected for the study were:” was replaced with "The samples were tested by RT-PCR/PCR based assays for the following viral species:"
L24: the phrase "They were analyzed by RT-PCR using specific primers for each one." was deleted
L28: "first report" was changed to "first field survey report"
L30: "the knowledge of the sanitary status in commercial vineyards" was change to "the knowledge to maintain the sanitary status of commercial vineyards"
L35: the reviewer stated: 8th what? in terms of acreage or wine production. The phase ”Argentina is the 8th grapevine producer worldwide” was replaced with “Argentina is the 7th largest grapevine producer worldwide”
L37: the reviewer stated: "second wine region"....Is it second largest? be specific. The phrase was replaced with “second largest wine region”
L40: the reviewer stated: "red vines" not sure if this is right wording. The phrase was replaced with “red grapes”
L40-42: the sentence “In these valleys, red vines represent 66% of the surface with grapevines cultivars, prevailing the Malbec and Cabernet Sauvignon varieties, 32% are white vines and 2% pink vines.” was replaced with ”In these valleys, red grapes represent 66% of the surface planted with grapevines cultivars, with the Malbec and Cabernet Sauvignon being the prevailing varieties, whereas white grapes represent 32% and pink grapes 2%.”
L45: the reviewer stated: "stabilishing new ones?". The phrase “The trend is to continue increasing the cultivated area, expanding farms and stablishing new ones” was replaced with “The trend is that the cultivated area, as well as the expansion and establishment of vineyards, will continue to increase”.
L46: the reviewer stated:"planting materials (certified virus-free or not) of the" change to "planting material (certified virus-free or not) used for establishing vineyards in CV are".
The sentence was replaced with” Most of the planting materials (either certified virus-free or not) used to establish vineyards in the Calchaquíes Valleys are imported from nurseries located in the provinces of Mendoza and San Juan, in the Cuyo region of Argentina, with a smaller proportion coming from foreign nurseries or being generated by the producer themselves.”
L54: "treatments" was replaced with "methods"
L75: "to perform an" was replaced with "for"
L77: "to perform a first diagnosis of grapevines" was replaced with "to diagnose the spectrum of viruses present in vineyards"
L94: the reviewer stated: "presented twelve different combined infections" ???? rephrase. Use the wording "mixed infections" for co-infection of two or more viruses in one sample.
“combined infections” was replaced with “mixed infections”
L124: the phrase “In locations, where the presence of mealybugs was not detected, the prevalence of GLRaV-3 was also observed:” was replaced with "GLRaV-3 was also detected in localities where mealybugs were not found:"
L145: "vines samples" was changed to "vine samples"
L145: the phrase “Although GRBV was not detected in these vines samples, we cannot assure its absence in the region, considering it was previously reported in the country, in the province of Mendoza by Luna et al. [10].” was replaced with “Although GRBV was not detected in the vine samples studied, the absence of this virus in the region cannot be confirmed with certainty, considering it has been previously reported in the country, in the province of Mendoza, by Luna et al. [10].”
L157: the reviewer stated: “You may want to indicate even at the beginning of the results section that nine out of the 103 samples collected were asymptomatic. Also, indicate the viruses that were detected in these asymptomatic samples.” The phrase “Moreover, four out of the nine asymptomatic plants sampled in the present study tested positives for GFLV, GLRaV-4 and/or GVA, suggesting that asymptomatic plants could be a source of inoculum if used as propagation material.” was added.
L181: the reviewer stated: Indicate the time of the year (month) you collected the mealybug presence/ absence data.
The phrase “The presence/absence of mealybugs (P. ficus) was also recorded at the end of summer and the beginning of autumn (February and March respectively).” was added
L210: "second wine region" was replaced with “second largest wine region”
Figure 2: the legend was rephrased
"Virus and virus mixed infections detected in the grapevines from northwestern Argentina studied"
The reviewer stated: The primer sequences table (supplementary table S2) can be considered to move into main draft if it is allowed as per short communication instructions. Table S2 was inserted as Table2
Reviewer 2 Report
Rivadeneira and colleagues present a survey on major grapevine viruses collected in NW Argentinian vineyard. In total 103 samples were tested and the results are presented in a way that is scientifically common practice.
I do have one major concern: The authors please need to double-check their reesults on GRBV, because the primers
|
F: CAAGTCGTTGTAGATTGAGGACGTATTGG |
R: AGCCACACCTACACGCCTTGCTCATC
will not give an amplification product if cDNA (from RT reaction) is used.
Since GRBV is a DNA virus one should use a DNA prep as template or adjust the primers for mRNA / RT / cDNA.
I think this is the reason why no GRBV was detected at all, but maybe the authors prove me wrong here.
In any case I would like to suggest to improve figure S1. First put the map into the manuscript and include piecharts of the detected viruses into the map, so one can see how many samples were tested where and how many and which viruses were identified. So one has all information in one figure. Fgure 1 is the figure 2 or suppl. figure.
The manuscripts reads well, although minor corrections will be necessary to imrpove the scientific English, for example replace "et al." in the text with "and colleagues".
Nice work - if the GRBV has been double-checked!
Author Response
REVIEWER 2 stated: I do have one major concern: The authors please need to double-check their results on GRBV, because the primers
F: CAAGTCGTTGTAGATTGAGGACGTATTGG
R: AGCCACACCTACACGCCTTGCTCATC
will not give an amplification product if cDNA (from RT reaction) is used. Since GRBV is a DNA virus one should use a DNA prep as template or adjust the primers for mRNA / RT / cDNA. I think this is the reason why no GRBV was detected at all, but maybe the authors prove me wrong here.
Thanks for the observation, it is a pertinent question. The primers used for GRBaV diagnostics in this manuscript, originally described by Krenz et al 2014, are targeted to C1 ORF, coding for a replication associated protein. This ORF is transcribed into a sub genomic RNA, which could be reverse transcribed trough RT as indicated in the manuscript. This method for identification of GRBaV was explored previously by Xiao et al (Virol J 12:171), which demonstrate that total nucleic acids or total RNA extracts are well suited for DNA virus diagnostics. In order to support this, attached you will found as you requested a double check with a second primer pair into a limited number of samples. We perform a PCR over the cDNAs with the primers described in the manuscript (complementary figure 1), the primers GRBaV1097F/1331R (as described by Xiao et al, complementary figure2 ), and the primers targeted to V.v Actin described by Muñoz et al (J. Agric. Food Chem. 62: 6716–6725) (complementary figure3).
Reviewer 2 stated: I would like to suggest to improve figure S1. First put the map into the manuscript and include pie-charts of the detected viruses into the map, so one can see how many samples were tested where and how many and which viruses were identified. So one has all information in one figure. Figure 1 is the figure 2 or suppl. figure.
Figure S1 was improved and included in the manuscript as Figure 1 as suggested by the reviewer.
Reviewer 2 stated: The manuscripts reads well, although minor corrections will be necessary to improve the scientific English, for example replace "et al." in the text with "and colleagues".
“et al.” was replaced with “and colleagues” throughout the manuscript.

Round 2
Reviewer 1 Report
in Table #2 authors cited reference #26 for several of the primers used in the current study. Please verify if the cited reference is the primary source for these primers. If not I recommend authors to cite appropriate primary source of these primer sequences.
Author Response
Thank you for your observation. After revise the cited work, we incorporate the original source for GVA primers in the Table 2 and into the bibliography. Indeed, we rearrange the bibliography for an accurate sorting order.
Reviewer 2 Report
Authors did a good job. I was wrong. Manuscript can be recommended for publication. Figure is improved. All concerns addressed.
Happy Friday
Author Response
Thank you for all your suggestions, it really help to improve the manuscript.